# Correlation of the High-Resolution Computed Tomography Patterns of Intrathoracic Sarcoidosis with Serum Levels of SAA, CA 15.3, SP-D, and Other Biomarkers of Interstitial Lung Disease

**DOI:** 10.3390/ijms241310794

**Published:** 2023-06-28

**Authors:** Zala Leštan Ramovš, Snežna Sodin-Šemrl, Katja Lakota, Saša Čučnik, Damjan Manevski, Rok Zbačnik, Mirjana Zupančič, Martin Verbič, Marjeta Terčelj

**Affiliations:** 1Department of Pulmonary Diseases, University Medical Centre Ljubljana, Zaloška 7, 1000 Ljubljana, Slovenia; zala.lestan.ramovs@kclj.si; 2Faculty of Mathematics, Natural Sciences and Information Technologies (FAMNIT), University of Primorska, 6000 Koper, Slovenia; snezna.semrl@upr.si (S.S.-Š.); katja.lakota@kclj.si (K.L.); 3Department of Rheumatology, University Medical Centre Ljubljana, 1000 Ljubljana, Slovenia; sasa.cucnik@kclj.si; 4Faculty of Pharmacy, University of Ljubljana, 1000 Ljubljana, Slovenia; 5Institute for Biostatistics and Medical Informatics, Faculty of Medicine, University of Ljubljana, 1000 Ljubljana, Slovenia; damjan.manevski@mf.uni-lj.si; 6Institute of Radiology, University Medical Centre Ljubljana, Zaloška 7, 1000 Ljubljana, Slovenia; rok.zbacnik@kclj.si; 7Laboratory Department, Children’s Hospital, University Medical Center Ljubljana, Zaloška 7, 1000 Ljubljana, Slovenia; mirjana.zupancic@kclj.si; 8Faculty of Medicine, University of Ljubljana, 1000 Ljubljana, Slovenia; martin.verbic@outlook.com

**Keywords:** intrathoracic sarcoidosis, biomarker, HRCT, Scadding stage

## Abstract

Studies on the serum biomarkers of granulomatous inflammation and pulmonary interstitial disease in intrathoracic sarcoidosis have shown conflicting results. We postulated that differences in the concentrations of serum biomarkers can be explained by the heterogenous patterns of sarcoidosis seen on thoracic HRCT. Serum biomarker levels in 79 consecutive patients, newly diagnosed with intrathoracic sarcoidosis, were compared to our control group of 56 healthy blood donors. An analysis was performed with respect to HRCT characteristics (the presence of lymph node enlargement, perilymphatic or peribronchovascular infiltrates, ground-glass lesions, or fibrosis), CXR, and disease extent. Serum levels of CXCL9, CXCL10, CTO, and CCL18 were statistically significantly increased in all patients compared to controls. Serum levels of CA15.3 were statistically significantly increased in all patients with parenchymal involvement. SAA was increased in patients with ground-glass lesions while SP-D levels were statistically significantly increased in patients with lung fibrosis. Only SP-D and CA15.3 showed a significant correlation to interstitial disease extent. In conclusion, we found that sarcoidosis patients with different HRCT patterns of intrathoracic sarcoidosis have underlying biochemical differences in their serum biomarkers transcending Scadding stages. The stratification of patients based on both radiologic and biochemical characteristics could enable more homogenous patient selection for further prognostic studies.

## 1. Introduction

Sarcoidosis is a rare, notoriously heterogenous multisystem inflammatory disorder of unknown aetiology [1]. It mostly affects young adults between 30 and 50 years of age and is characterized by the formation of non-caseous granulomas in affected organs [2,3,4]. The intrathoracic lymph nodes and lungs are affected in 90% of patients, with cough, chest pain, and dyspnoea upon exertion amongst the most common symptoms [5]. Positive diagnosis requires histological confirmation of non-necrotising granulomas at disease sites in patients with a compatible clinical and radiological presentation and after careful exclusion of other possible differential diagnoses [4,6]. Clinical manifestation varies from asymptomatic, to acute febrile, to chronic and highly debilitating [7]. The outcome of intrathoracic sarcoidosis is unpredictable, with two thirds of patients achieving spontaneous remission, the majority within one to three years of diagnosis. However, the remaining third of patients suffer a chronic progressive or relapsing-remitting disease course, necessitating immunosuppressive treatment. Approximately 10–20% of patients with intrathoracic involvement eventually develop irreversible lung damage, with loss of diffusion capacity for carbon monoxide, progressive fibrosis, and loss of lung volume, eventually leading to respiratory insufficiency and possible pulmonary hypertension [8,9,10].

The heterogenicity of sarcoidosis complicates decision making regarding treatment and follow-up, necessitating further research into disease subtypes and the prognostic features of sarcoidosis [11,12]. Considerable effort has been made to identify serum biomarkers that can assist in the detection of active and chronic progressive disease [13,14]. Amongst the most widely studied are biomarkers with high diagnostic sensitivity, such as chitotriosidase (CTO) (sensitivity 89%, specificity 93%) [15] and the acute phase reactant serum amyloid A (SAA) (sensitivity rate 96%, specificity 37–52%) [16,17,18]. Th1/Th17 disease biomarkers include chemokines CXCL9 (monokine induced by interferon-γ) and CXCL10 (interferon-γ-induced protein 10) [13,19]. Potential prognostic value has been demonstrated for surfactant protein D (SP-D), CC chemokine ligand 18 (CCL18), and cancer antigen 15.3 (CA 15.3)/Krebs von den Lungen 6 (KL-6) acting as biomarkers for progressive/fibrosing disease [20]. Clinical utility remains equivocal; however, we postulate that the inconsistency in study results can be explained, at least in part, by the heterogenous patterns of lung involvement in sarcoidosis, as evidenced on thoracic high-resolution computed tomography (HRCT).

The rationale and goal of our study was to confirm whether characteristic HRCT patterns of sarcoidosis correlate to differences observed in the serum levels of SAA and the serum markers of granulomatous inflammation and pulmonary interstitial disease, and thus could yield additional and/or better support than traditional chest X-ray (CXR)-based Scadding disease stages [21]. This could then justify the stratification of patients based on both radiologic and biochemical characteristics, resulting in a more optimal and homogenous patient selection for further prognostic studies.

## 2. Results

Patient demographic features, along with pulmonary function test values, BAL CD4+/CD8+ ratio, and HRCT patterns, are reported in Table 1 and Table 2. The onset of disease occurred mostly in the fourth to sixth decade of age, mainly in females who had never smoked. The majority of patients had preserved lung volumes; however, there was a mild reduction in median diffusion capacity for carbon monoxide (DLCO) (80%). There was no statistically significant difference in spirometry values between radiologic phenotypes. All patients have CXR evidence of intrathoracic disease (Scadding stage 0, *n* = 0), with most in Scadding stage 2 (*n* = 63). Only one patient is in Scadding stage 4 (Table 3). Of the 79 patients, 76 had a thoracic HRCT or PET/CT performed at the time of diagnosis. At disease onset, hilar or mediastinal lymph node involvement was present in 66 patients, perilymphatic nodules in 42, peribronchovascular infiltrates in 33, ground-glass infiltrates in 14, and lung fibrosis in 11 patients (images included as Appendix A). Based on a multidisciplinary review and clinical, radiological, and histological correlation according to ERS/WASOG [4,6], 55 patients were determined as having a definite diagnosis of sarcoidosis interstitial lung disease.

The focus of this report was to compare the serum levels of SAA and the biomarkers of granulomatous inflammation and pulmonary interstitial disease in sarcoidosis patients to our control group with respect to typical HRCT patterns of sarcoidosis (Table 4). When comparing the entire sarcoidosis cohort with the healthy cohort, the values for all markers (SAA, CCL18, CA15.3, CXCL9, CXCL10, and CTO) statistically significantly differ (*p*-values ≤ 0.001), except for SP-D (*p*-value = 0.23).

The serum levels of SAA are statistically significantly increased in sarcoidosis patients compared to controls (*p* = 0.00–0.021); however, only in patients with ground-glass lesions do the median serum levels go above normal (median value 9 μg/mL, with a cut-off for physiological concentrations of 6.4 μg/mL). SP-D levels are increased and statistically significant in patients with lung fibrosis (*p* = 0.014, median of 30,178 pg/mL vs. 11,901 pg/mL in healthy controls) followed by ground-glass lesions on HRCT (*p* = 0.059, with a median of 16,639 pg/mL). Statistically significant results are also obtained regarding levels of CA15.3 (*p* = 0.006–0.029, median values ranging from 38 pg/mL to 47 pg/mL in groups of patients divided by HRCT pattern vs. 29 pg/mL in healthy controls).

The values of the serum analyte levels in patients with specific HRCT disease patterns and healthy controls are represented with box plots in Figure 1. While the serum levels of CCL18, CXCL9, and CXCL10 are similarly increased regardless of HRCT patterns, more heterogenous values are observed for SAA, SP-D, and CA15.3, most notably in the case of SP-D.

For comparison, Figure 2 shows the distribution and median values of the serum analyte levels in CXR Scadding stages compared to healthy controls. The sarcoidosis cohort has a numerically inhomogeneous distribution according to Scadding stages with most patients in stage 2, while stage 3 only has 6 patients and only 9 patients are in stage 1. There is only one patient in Scadding stage 4, and no conclusions could be drawn. Scadding stage 0 is omitted, as there were no patients in this group.

Serum levels of CCL18, CXCL9, and CXCL10 are statistically significantly increased in all patients, regardless of Scadding stages (*p* values ranging from <0.001 to 0.035). Serum levels of SAA are increased in patients in Scadding stage 1 compared to controls (with a median equal to 7.13 μg/mL, *p*-value = 0.003). In Scadding stage 2, the median concentration of SAA does not exceed the cut-off for physiological concentrations of 6.4 μg/mL. The highest concentrations of SP-D are observed in Scadding stage 3 with a median equal to 20,330 pg/mL, *p*-value = 0.14. CA15.3 is likewise increased and borderline statistically significant only in stage 3 disease (median equal to 50.46, *p*-value = 0.07).

Next, the influence of the extent of interstitial lung disease on the serum levels of selected markers was determined. There were 21 patients with no interstitial lung disease, 26 with minor interstitial lung disease, 20 with moderate interstitial lung disease, and 9 with severe interstitial lung disease. Only CA15.3 and SP-D showed a statistically significant correlation with the radiologic extent of pulmonary sarcoidosis (correlation coefficients: 0.30 and 0.37 and *p*-values: 0.03 and 0.01, respectively). The data are shown in Figure 3.

CTO has a set maximum reference value in healthy subjects in our laboratory, enabling CTO data to be gathered only for sarcoidosis and allowing comparison with the null value of CTO = 65 nmol/h/mL (the maximum reference value). The result was statistically significant (*p*-value < 0.001), with the median CTO equal to 840 nmol/h/mL (IQR: (390, 1320)). CTO was considered with respect to HRCT patterns (the distribution is shown on the left graph in Figure 4). CTO was similarly distributed across all HRCT characteristics, except in ground-glass lesions, where, on average, larger CTO values were observed. However, a Kruskal–Wallis test was performed to compare the HRCT characteristics and the result was not statistically significant (*p*-value = 0.08). On further analysis, no statistically significant relationship was found between CTO and either Scadding score (middle graph in Figure 4) or HRCT disease extent (right graph in Figure 4; correlation coefficient 0.16, *p*-value = 0.16).

## 3. Discussion

The results of our study show underlying biochemical differences between four typical HRCT patterns of intrathoracic sarcoidosis, namely: predominant hilar and mediastinal lymph node enlargement, perilymphatic nodules and/or peribronchovascular infiltrates, ground-glass lesions, and pulmonary fibrosis. All our sarcoidosis patients had increased serum concentrations of CXCL9, CXCL10, CTO, and CCL18. However, patients with either perilymphatic nodules and/or peribronchovascular infiltrates additionally exhibited increased serum levels of CA15.3. In the presence of ground-glass lesions, elevated levels of both CA15.3 and SAA were prominent, while HRCT evidence of pulmonary fibrosis was accompanied by an increase in both CA15.3 and SP-D

Imaging studies have long been valued as both diagnostic and prognostic tools in intrathoracic sarcoidosis [21,22,23]. However, assessment continues to rely on the 1961 Scadding staging system, based on CXR [21,23]. Intrathoracic sarcoidosis has traditionally been classified into 5 Scadding stages (illustrated in Figure 5) that do not reflect the natural progression of sarcoidosis but rather the decreasing likelihood of spontaneous disease remission [21]. The Scadding system serves well when identifying extremes of sarcoidosis. However, HRCT allows earlier detection and differentiation of parenchymal lesions while achieving better interobserver agreement than CXR [24]. In our study, HRCT is shown to be more sensitive than CXR for discrete interstitial abnormalities, especially in Scadding stage 1. It allows for better visualization and categorization of interstitial disease involvement. Furthermore, HRCT successfully identifies 11 patients with pulmonary fibrosis (as opposed to the one ascribed to Scadding stage 4 by CXR analysis).

Moving on to biomarker analysis, our sarcoidosis cohort displays statistically significantly increased levels of granulomatous disease markers CXCL9 and CXCL10. CXCL9 and CXCL10 are chemokines secreted by alveolar macrophages, multinucleated giant cells, and epithelioid histiocytes that bind to the CXCR3 receptor and recruit CD4+ T cells, monocytes, and other inflammatory cells to the site of inflammation. In sarcoidosis patients, their elevated levels reflect the ongoing Th1 inflammatory response [14].

CCL18 is a chemokine produced by antigen-presenting cells that is chemotactic for both naive and activated T lymphocytes and has been shown to upregulate both mRNA and protein production of collagen in lung fibroblasts in vitro [25]. In vivo, however, CCL18-mediated inflammation and fibrosis is dependent on T lymphocyte infiltration and more complex [26]. Accordingly, while studies have shown that CCL18 plays a role in fibrotic interstitial lung disease [25,27], higher CCL18 levels have been observed in lung diseases with a prominent lymphocytic inflammatory component, such as sarcoidosis [28]. In our sarcoidosis cohort, CCL18 levels are increased in all patient groups, with a peak in fibrotic disease (with a median of 66,812 pg/mL vs. 59,317 pg/mL in the entire sarcoidosis cohort).

CTO is a highly sensitive biomarker of sarcoidosis secreted by activated macrophages and neutrophils [29]. It most likely reflects the granulomatous burden of the disease in terms of sarcoidosis activity and severity and has been shown to predict clinical course, steroid responsiveness, and disease relapse [30]. Although we have not directly compared CTO activity in sarcoidosis patients and healthy controls in this study, the results are in accordance with previously published research performed by our sarcoidosis group in the same population where direct comparison of patients and healthy controls showed significantly increased CTO activity in sarcoidosis patients (*p* < 0.001), confirming the high sensitivity of this disease marker [31,32,33]. CTO activity strongly correlated with disease activity and severity and was confirmed to be useful in patient follow-up and detection of disease relapse [30,34]. Despite having good specificity, activity above controls was also found in patients with asbestosis, non-sarcoidosis pulmonary fibrosis, and lung cancer [31]. As expected, CTO activity is statistically significantly increased in our sarcoidosis cohort; however, there was no statistically significant relationship with either Scadding stage or HRCT disease extent. CTO activity was similar, regardless of HRCT patterns, except in ground-glass lesions, although the *p*-value remains non-significant (0.08).

CA 15.3/KL-6 is expressed on Type II pneumocytes and respiratory bronchiolar epithelial cells. It has been studied as a marker of interstitial lung disease, as it promotes the chemotaxis of human fibroblasts, accelerates their proliferation, and inhibits apoptosis of human lung fibroblasts [35,36]. In sarcoidosis, KL-6 corresponds to lymphocytic alveolitis and is associated with increased parenchymal infiltration [37]. In our sarcoidosis cohort, CA15.3 is statistically significantly increased in all patients with parenchymal lung involvement. Serum levels of CA15.3/KL-6 are highest in patients with ground-glass lesions. While in our study, statistical significance in not achieved, previous studies have demonstrated a statistically significant relationship between the serum levels of KL-6 and ground-glass opacities, corresponding to active disease on 67-gallium scintigraphy and a higher CD4+/CD8+ BAL ratio [38]. In our study, serum concentrations of CA15.3 showed a statistically significant correlation with the radiologic extent of disease, similar to the results of a previous study demonstrating a correlation of KL-6 with the severity of fibrotic lung involvement and functional impairment [20]. Among patients, the Scadding stage is not significantly correlated with CA15.3 (correlation coefficient 0.13, *p* value 0.26).

Patients with ground-glass lesions are a seemingly biochemically specific subgroup of intrathoracic sarcoidosis, with increased levels of SAA, CXCL9, CXCL10, CCL18, CTO, and CA15.3. Histologically, ground-glass lesions correlate with coalescent micronodules of granulomatous inflammation [10]. Research has shown that SAA could have a role in the formulation and perpetuation of the granulomatous response in sarcoidosis. SAA activates NF-kappaB and triggers a Th1 immune response with an increased production of IFN-γ, TNF-α, IL-10, and IL-18 via its interaction with the Toll-like receptor 2 [16]. Increased serum levels of SAA in patients with ground-glass lesions can thus be correlated with active granuloma formation.

In patients with pulmonary fibrosis, increased levels of CXCL9, CXCL10, CCL18, CTO, and CA15.3 are accompanied by elevated SP-D, a component of lung surfactant and a pattern-recognition molecule produced by alveolar type II cells and Clara cells. The serum levels of SP-D are increased in patients with interstitial lung diseases, likely due to a combination of increased production by (regenerated) alveolar type II cells and increased leakage caused by enhanced permeability [39]. It has been reported that SP-D correlates with the severity of interstitial lung disease [40] and in our study, there is a statistically significant correlation between SP-D concentrations and disease extent. Other studies have demonstrated possible prognostic implications, with serum concentrations of SP-D corresponding to worsening DLCO% and HRCT evidence of progression of sarcoidosis pulmonary fibrosis [41].

Patients with peribronchovascular and perilymphatic nodules share a common biochemical milleu, with increased levels of granulomatous disease markers CXCL9, CXCL10, CCL18, and CTO and increased CA15.3. Kobayashi et al. also found a statistically significant relationship between the serum levels of KL-6 and a thickened peribronchovascular bundle [38].

In patients with HRCT evidence of only hilar or mediastinal lymph node enlargement, biomarkers reflect a Th1 disease process, with increased CXCL9, CXCL10, CCL18, and CTO. While the serum levels of CA15.3. are significantly increased in patients with lymph node enlargement compared to healthy controls, this possibly reflects simultaneous interstitial involvement, as an analysis by Scadding stage shows virtually no difference in the median value of CA15.3 between healthy controls and patients with Scadding stage 1.

As all studies, ours has strengths and also inherent weaknesses. Recently, we have seen a shift away from CXR-based Scadding staging of sarcoidosis with clinicians relying more and more on HRCT for diagnostic and treatment decisions [24,42]. However, studies based on HRCT are far from abundant. Some authors have concentrated on composite HRCT scores in their search for prognostic information [22,43]. To our knowledge, ours is the first study found in the literature focusing on commonly reported HRCT characteristics and their relationship with biomarkers of granulomatous inflammation and pulmonary interstitial fibrosis in sarcoidosis. It was further expanded by analyzing the relationship between serum biomarkers and disease extent. A comparison with Scadding stage was performed and discussed.

This is a referral-center study on newly diagnosed, treatment-naïve sarcoidosis patients that benefited from extensive multidisciplinary involvement of specialists in pneumology, thoracic radiology, pathology, and biochemistry. All clinical workup from history-taking to bronchoscopy was performed by two physicians dedicated to sarcoidosis. Two specialists in thoracic radiology separately interpreted each HRCT and CT from 18-FDG-PET/CT. Serum samples were frozen and stored and analysis performed concurrently for the whole group, avoiding bias.

However, our study has some limitations. Our patient group features an uneven distribution of patients according to Scadding stage. Scadding stage 0 was purposely omitted, as we focused on patients with confirmed intrathoracic disease. While we would expect most patients (45–65%) to be in Scadding stage 1, this number is much lower in our patient group, only 11.5%, with 79.7% of patients in stage 2 and 7.6% in stage 3 [9]. In patients with stage 1 sarcoidosis that were referred to our sarcoidosis centre and presented with Löfgren syndrome, HRCT and histopathologic confirmation of diagnosis were only rarely performed due to the diagnostic features of Löfgren syndrome and the mostly benign disease course [6,44]. In contrast, both HRCT and histopathologic confirmation of disease by bronchoscopy were needed for inclusion in our study. Additionally, some patients with Löfgren syndrome, as a form of stage 1 sarcoidosis, were diagnosed by specialists in rheumatology or dermatology, who also initiated treatment before referral, again precluding their inclusion in our study, which only enrolled treatment-naïve patients.

While the number of patients is respectable for a single-center study on new-onset sarcoidosis, statistical analysis was hindered by the low frequency of outcomes and many possible confounding variables. Genetic differences have not been accounted for. Additionally, there is rarely only one prevailing HRCT pattern, making it difficult to discriminate the disease burden contributed by one specific radiologic variable.

## 4. Materials and Methods

### 4.1. Participants

Patients above 18 years of age were enrolled immediately after the first diagnostic workup. Patients ranged from 28.5 to 75.8 years of age, with the majority in the fourth to sixth decade (median age was 46.7). Most patients were female (58.3%). The duration of symptoms before diagnosis was at minimum a few days (mainly in patients with Löfgren syndrome characterized by fever, erythema nodosum, and bilateral hilar lymphadenopathy) and up to three years, with a median of three months. Symptoms included cough, chest pain, arthralgia, erythema nodosum, dyspnoea upon exertion, febrile temperatures, night sweats, decreased appetite, and fatigue. All patients underwent complete functional and radiological assessment within 2–4 weeks from referral and complied with the accepted Respiratory Society/World Association of Sarcoidosis and Other Granulomatous (ERS/WASOG) disease guidelines for the diagnosis of sarcoidosis [4,6]. Histological confirmation of non-caseating granulomas was obtained via flexible bronchoscopy, as well as bronchoalveolar lavage (BAL) fluid for the determination of BAL CD4+/CD8+ ratio. Patients with other known systemic inflammatory illnesses were excluded, as well as patients with acute infection, patients on immunosuppressive drugs or immunotherapy, and patients with active cancer. Radiological, histopathological, and biochemical analysis was performed before any treatment was initiated [4].

For a control group, a random sample of 56 healthy donors (27 female, 29 male), median age 43 (interquartile range (IQR): (34, 51)) was obtained from the Blood Transfusion Centre of Slovenia. The donors’ smoking status and comorbidities remained anonymous.

All patients provided informed written consent for participation in the study, which was approved by the Medical Ethics Committee of the Republic of Slovenia (#0120-77/2018/2). The study was registered at ClinicalTrials.gov (NCT05811962).

### 4.2. Chest Radiograph in Two Projections and Staging by Scadding

At the time of diagnosis, a CXR was taken in the anteroposterior and lateral projections using a Siemens Polydoros IT. The Scadding radiological stage was determined as follows: 0—no disease, 1—hilar or mediastinal nodal enlargement, 2—hilar or mediastinal nodal enlargement and parenchymal disease, 3—parenchymal disease only, and 4—lung fibrosis and volume loss [21].

### 4.3. Thoracic HRCT

Thoracic HRCT was performed at diagnosis and interpreted by a specialist of thoracic radiology, according to the standards of interpreting interstitial lung disease. For our study, a specialist of thoracic radiology from the Clinical Institute of Radiology, University Medical Centre, Ljubljana, performed an additional post-hoc blinded analysis of the HRCT scans. Alternatively, when no thoracic HRCT was available, low-dose CT from 18-fluorodeoxyglucose-positron emission tomography (18-FDG-PET) was interpreted. Patients were given yes/no scores in the following categories of HRCT lesions, commonly found in sarcoidosis: a. hilar or mediastinal lymph node enlargement, b. perilymphatic nodules (adjacent to interlobular septa, interlobar fissures, or subpleural nodules), c. peribronchovascular infiltrates, d. ground-glass lesions, and e. fibrosis [9,45] (characteristic HRCT images provided in Appendix A).

Additionally, based on the extent of disease, patients were grouped into the following descriptive categories: 0—no interstitial lung disease, 1—minor interstitial lung disease (0–33% of lung parenchyma involved), 2—moderate interstitial lung disease (33–66% of lung parenchyma involved), and 3—severe interstitial lung disease (more than 66% of lung parenchyma involved) [46,47].

Where there was a divergence between the interpretation of the two radiologists, an opinion of a clinician specializing in sarcoidosis was sought.

### 4.4. Serum Samples

Serum samples were collected after clotting of peripheral venous blood and centrifuged at 1800× *g* for 10 minutes, aliquoted, and stored short-term at −20 °C and long-term at −80 °C until used for further analysis.

#### 4.4.1. Multiplex Biomarker Analysis Procedure

The detection and quantification of multiplex biomarkers in sera was conducted using Magnetic Luminex Assay (R&D, Abingdon, UK) with MagPix (Luminex, Austin, TX, USA), according to manufacturer’s instructions. A 4-parametric standard curve analysis was used, and target analyte concentrations were determined in the samples.

#### 4.4.2. Other Measurements

SAA was quantified in sera using particle immunonephelometry with a BN Prospec system (Siemens, Marburg, Germany), according to the manufacturer’s instructions. The result was evaluated by comparison with a known standard concentration. The CTO activity was determined in sera using the 22 μM 4-methylumbelliferyl-β-D-N,N′,N″-triacetylchitotriosiose (4 MU-chitotrioside, Sigma-Aldrich Chemical Co., St. Louis, MO, USA) in citrate phosphate buffer (pH 5.2) as an enzymatical substrate. The reaction product, fluorescent 4-methylumbelliferone, was measured using a Perkin–Elmer fluorimeter (Perkin-Elmer Life and Analytical Sciences Inc., Wellesley, MA, USA) at excitation wave length 365 nm and emission 465 nm.

### 4.5. Statistical Analysis

All statistical analysis was performed using the R statistical software, version 4.0.3. [48].

In the article, we reported frequencies, medians, and the corresponding IQR (interquartile ranges). Spearman correlations and the corresponding correlation tests were reported as well. Comparisons with respect to HRCT phenotype at onset (lymph node enlargement, perilymphatic nodules, peribronchovascular infiltrates, ground-glass lesions, and fibrosis) were completed using the Wilcoxon–Mann–Whitney and Wilcoxon signed ranked tests. Where needed, *p*-values were corrected using the Benjamini and Hochberg method. The relationship between CTO and HRCT patterns was determined using the Kruskall–Wallis test.

## 5. Conclusions

In sarcoidosis, both the burden of disease and associated immunosuppressive treatment can be significant, emphasizing the importance of better understanding this multifaceted disease and identifying reliable indicators of clinical outcome. However, in an age focused on the phenotypisation of diseases, radiologic stratification of sarcoidosis continues to rely on CXR Scadding staging. Our knowledge of the underlying differences leading to the diverse radiological presentation of sarcoidosis on HRCT and especially of their possible prognostic implications remains incomplete. In this article, we have shown that routinely reported HRCT characteristics of sarcoidosis demonstrate underlying differences in the serum markers of granulomatous inflammation and pulmonary interstitial disease. Out of the typical HRCT characteristics of intrathoracic sarcoidosis, hilar and mediastinal lymph node enlargement, perilymphatic/peribronchovascular involvement, predominantly ground-glass lesions, and pulmonary fibrosis have shown differences in the measured serum markers, especially SAA, CA15.3, and SP-D. Further studies will be needed to establish the prognostic value and possible clinical implications of patient stratification according to our proposed HRCT disease patterns. One such follow-up study is already underway at our Sarcoidosis Centre at the University Medical Centre, Ljubljana.

## Figures and Tables

**Figure 1 ijms-24-10794-f001:**
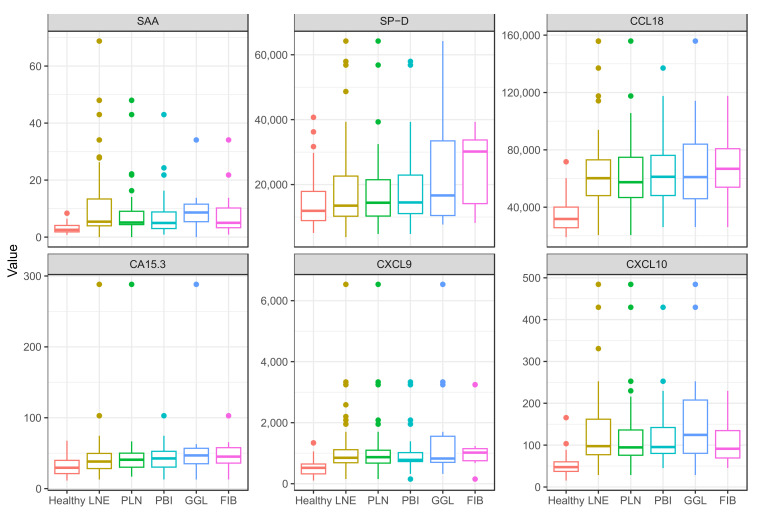
Comparison of serum levels of SAA, SP-D, CCL18, CA15.3, CXCL9, and CXCL10 in characteristic HRCT patterns of intrathoracic sarcoidosis and healthy controls. LNE: lymph node enlargement (*n* = 66), PLN: perilymphatic nodules (*n* = 42), PBI: peribronchovascular infiltrates (*n* = 33), GGL: ground-glass lesions (*n* = 14), FIB: fibrosis (*n* = 11), HRCT: high-resolution computed tomography, SAA: serum amyloid A, SP-D: surfactant protein D, CCL18: CC chemokine ligand 18, CA 15.3: cancer antigen 15.3, CXCL9: monokine induced by interferon-γ, CXCL10: interferon-γ-induced protein 10.

**Figure 2 ijms-24-10794-f002:**
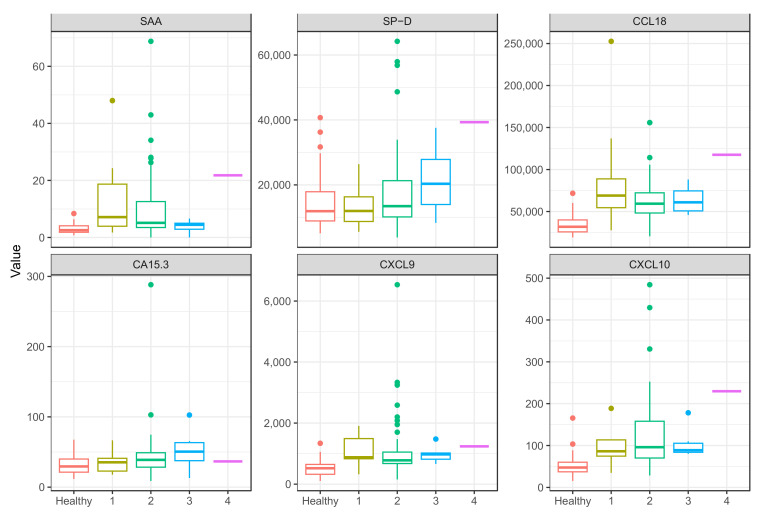
Comparison of serum levels of SAA, SP-D, CCL18, CA15.3, CXCL9, and CXCL10 in healthy controls and sarcoidosis patients in different Scadding stages. Scadding stage 0 was omitted as all patients have confirmed intrathoracic sarcoidosis. 1: Scadding stage 1 (*n* = 9), 2: Scadding stage 2 (*n* = 63), 3: Scadding stage 3 (*n* = 6), 4: Scadding stage 4 (*n* = 1), SAA: serum amyloid A, SP-D: surfactant protein D, CCL18: CC chemokine ligand 18, CA 15.3: cancer antigen 15.3, CXCL9: monokine induced by interferon-γ, CXCL10: interferon-γ-induced protein 10.

**Figure 3 ijms-24-10794-f003:**
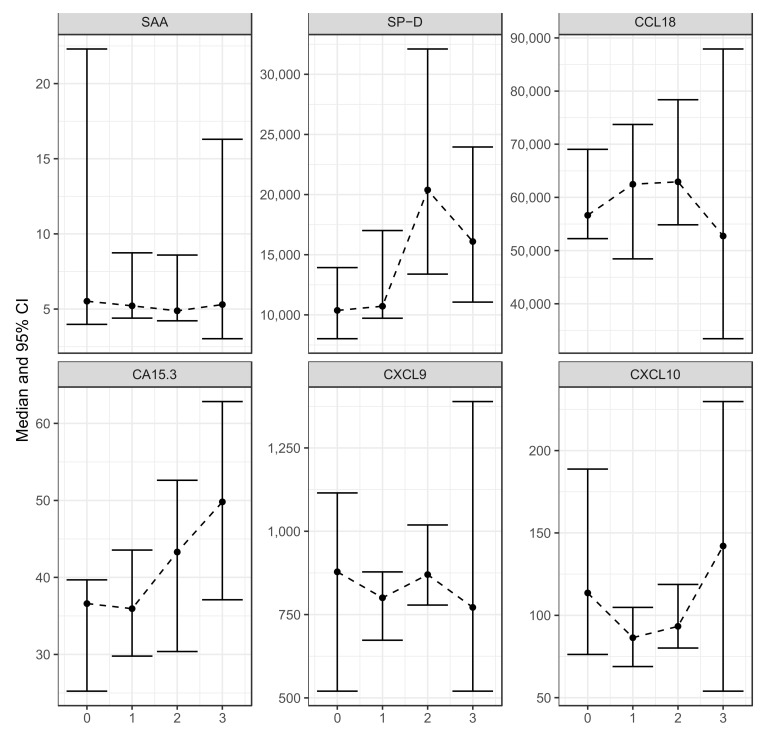
Comparison of the serum levels of SAA, SP-D, CCL18, CA15.3, CXCL9, and CXCL10 in patients with different extents of lung involvement. 0: no interstitial lung disease (*n* = 21), 1: minor interstitial lung disease with 0–33% of lung parenchyma involved (*n* = 26), 2: moderate interstitial lung disease with 33–66% of lung parenchyma involved (20), 3: severe interstitial lung disease with more than 66% of lung parenchyma involved (*n* = 9), SAA: serum amyloid A, SP-D: surfactant protein D, CCL18: CC chemokine ligand 18, CA 15.3: cancer antigen 15.3, CXCL9: monokine induced by interferon-γ, CXCL10: interferon-γ-induced protein 10. The central dots represent the median and the boxes represent the 95% confidence interval for the median.

**Figure 4 ijms-24-10794-f004:**
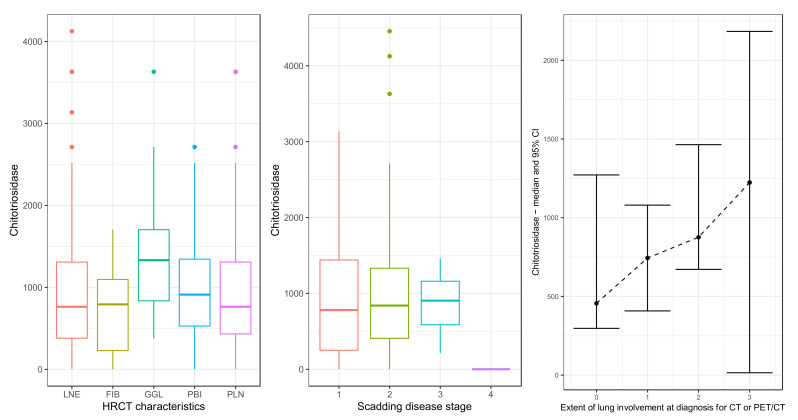
Relationship of chitotriosidase activity with HRCT patterns (**left** figure), Scadding stage (**middle** figure), and HRCT disease extent (**right** figure). HRCT characteristics (**left** figure): LNE: lymph node enlargement (*n* = 66), PLN: perilymphatic nodules (*n* = 42), PBI: peribronchovascular infiltrates (*n* = 33), GGL: ground-glass lesions (*n* = 14), FIB: fibrosis (*n* = 11). Scadding disease stage (**middle** figure): 1: Scadding stage 1 (*n* = 9), 2: Scadding stage 2 (*n* = 63), 3: Scadding stage 3 (*n* = 6), 4: Scadding stage 4 (*n* = 1). Extent of lung involvement at diagnosis for CT or PET/CT (**right** figure): 0: no interstitial lung disease (*n* = 21), 1: minor interstitial lung disease with 0–33% of lung parenchyma involved (*n* = 26), 2: moderate interstitial lung disease with 33–66% of lung parenchyma involved (20), 3: severe interstitial lung disease with more than 66% of lung parenchyma involved (*n* = 9), HRCT: high resolution computed tomography. The central dots represent the median; the tails and boxes represent the 95% confidence interval for the median.

**Figure 5 ijms-24-10794-f005:**
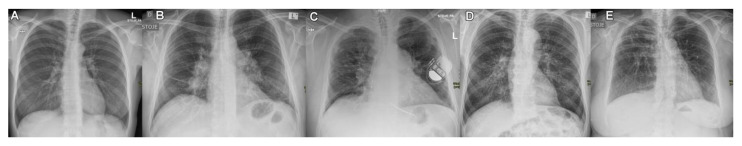
Scadding stages of intrathoracic sarcoidosis: (**A**): stage 0—no disease, (**B**): stage 1—hilar or mediastinal nodal enlargement, (**C**): stage 2—hilar or mediastinal nodal enlargement and parenchymal disease, (**D**): stage 3—parenchymal disease only, (**E**): stage 4—lung fibrosis and volume loss.

**Table 1 ijms-24-10794-t001:** Patient demographic and functional and BAL characteristics at diagnosis.

Variable	Patients (N = 79)
Gender	
Male	33
Female	46
Median age (IQR)	46.7 (38.2–55.6)
Smoking	
Never	60
Former smoker	10
Current smoker	9
Pulmonary function tests:	
-Median FVC in % (IQR)	96.0 (88.0–106.5)
-Median FEV1 in % (IQR)	94.5 (85.3–103.0)
-Median DLCO in % (IQR)	80 (73–89)
BAL CD4+/CB8+ cell ratio	4.84 (2.2–8.78)

BAL: bronchoalveolar lavage, IQR: interquartile range, FVC: forced vital capacity, FEV1: forced expiratory volume in 1 s, DLCO: diffusion capacity for carbon monoxide.

**Table 2 ijms-24-10794-t002:** Patient radiologic characteristics at diagnosis.

Variable	Patients (N = 79)
Scadding stage	
0	0
1	9
2	63
3	6
4	1
HRCT characteristics at diagnosis	44
-Lymph node enlargement	
Yes	38
No	6
-Perilymphatic nodules	
Yes	28
No	16
-Peribronchovascular infiltrates	
Yes	22
No	22
-Ground-glass lesions	
Yes	10
No	34
-Fibrosis	
Yes	7
No	37
18-FDG-PET/CT characteristics at diagnosis	32
(only patients with no HRCT at diagnosis)	
-Lymph node involvement	
Yes	28
No	4
-Perilymphatic nodules	
Yes	14
No	18
-Peribronchovascular infiltrates	
Yes	11
No	21
-Ground-glass lesions	
Yes	4
No	28
-Fibrosis	
Yes	4
No	28

HRCT: high-resolution computed tomography, 18-FDG-PET/CT: 18-fluorodeoxyglucose-positron emission tomography/computed tomography.

**Table 3 ijms-24-10794-t003:** Sarcoidosis CXR stages according to Scadding [21].

Scadding Stages	0	1	2	3	4
CXR findings	Normal	Hilar or mediastinal nodal enlargement	Hilar or mediastinal nodal enlargement and parenchymal disease	Parenchymal disease only	Lung fibrosis and volume loss
Number of patients	0	9	63	6	1

CXR: Chest X-ray.

**Table 4 ijms-24-10794-t004:** A comparison of levels of serum markers SAA, SP-D, CCL18, CA15.3, CXCL9, and CXCL10 between healthy controls and intrathoracic sarcoidosis patients with different CT patterns at diagnosis.

Variable	*n*	SAA [mcg/mL]	SP-D [pg/mL]	CCL18 [pg/mL]	CA15.3 [pg/mL]	CXCL9 [pg/mL]	CXCL10 [pg/mL]
Healthy cohort	56	3 (2; 4)	11,901 (8880; 17,878)	31,809 (25,711; 40,035)	29 (21; 40)	520 (321; 646)	48 (37; 60)
Sarcoidosis cohort	79	5 (4; 13)	13,533 (10,134; 21,292)	62,690 (48,298; 75,741)	38 (28; 49)	829 (673; 1112)	95 (73; 146)
*p*-value		0	0.23	0	0.003	0	0
Lymph node enlargement	66	5 (4; 13)	13,496 (10,242; 22,606)	60,258 (48,052; 72,987)	38 (28; 50)	850 (686; 1113)	98 (77; 162)
*p*-value		0	0.141	0	0.006	0	0
Perilymphatic nodules	42	5 (4; 9)	14,388 (10,311; 21,492)	57,413 (46,745; 74,788)	41 (30; 50)	870 (673; 1095)	95 (76; 136)
*p*-value		0	0.098	0	0.005	0	0
Peribroncho-vascular infiltrates	33	5 (3; 9)	14,477 (11,062; 22,911)	61,200 (48,140; 76,133)	43 (30; 53)	779 (727; 1019)	95 (80; 142)
*p*-value		0	0.092	0	0.005	0	0
Ground-glass lesions	14	9 (5; 12)	16,639 (10,457; 33,454)	61,003 (48,140; 76,133)	47 (35; 57)	824 (698; 1553)	124 (80; 208)
*p*-value		0	0.059	0	0.012	0	0
Fibrosis	11	5 (3; 10)	30,178 (14,122; 33,752)	66,812 (53,949; 80,782)	45 (36; 58)	1019 (753; 1147)	91 (69; 135)
*p*-value		0.021	0.014	0	0.029	0	0

SAA: serum amyloid A, SP-D: surfactant protein D, CCL18: CC chemokine ligand 18, CA 15.3: cancer antigen 15.3, CXCL9: monokine induced by interferon-γ, CXCL10: interferon-γ-induced protein 10. The table contains the median and corresponding IQR for both groups and the *p*-value from a Wilcoxon–Mann–Whitney test. All *p*-values are corrected using the Benjamini and Hochberg method.

## Data Availability

The data presented in this study are available on request from the corresponding author. The data are not publicly available due to patient privacy restrictions.

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
