# Peer review of "Correlation of the High-Resolution Computed Tomography Patterns of Intrathoracic Sarcoidosis with Serum Levels of SAA, CA 15.3, SP-D, and Other Biomarkers of Interstitial Lung Disease"

_ijms, 2023, doi:10.3390/ijms241310794_

Round 1

Reviewer 1 Report

1.     Introduction 41-47:” Sarcoidosis is an heterogenous and unpredictable inflammatory multisystem disorder, characterised by the presence of noncaseous granulomas in affected organs. Intrathoracic sarcoidosis, involving hilar and mediastinal lymph nodes and the lungs is the most common manifestation, accounting for much of the disease's morbidity and most of its mortality. The disease course and outcome of intrathoracic sarcoidosis is variable, with two thirds of patients achieving spontaneous remission, while the other third suffer a chronic progressive or relapsing-remitting disease course. Approximately 20% of patients with intrathoracic involvement develop pulmonary fibrosis with loss of organ function.” Please improve this paragraph to explain this disease. I suggest some references such as:

-        J Clin Med. 2021 Jun; 10(11): 2462. doi: 10.3390/jcm10112462 Correlation between Potential Risk Factors and Pulmonary Embolism in Sarcoidosis Patients Timely Treated

-        DOI:
Risk factor of pulmonary embolism in sarcoidosis patients: a pilot study

2.     Results 73-75: “There was no statistically significant difference in spirometry 73 values between radiologic phenotypes. All patients have CXR evidence of intrathoracic disease (Scadding stage 0, n=0), 74 with most in Scadding stage 2 (n=63).” Please explain scadding stage, a simple table could be useful.

3.     In the discussion of the article, I would suggest sample tc images of the different stages of sarcoidosis.

4.         Partecipants 278-285: “Patients above 18 years of age were enrolled immediately after the first diagnostic workup and before any treatment was initiated. All patients have undergone complete functional and radiological assessment and comply with the accepted Respiratory Society/ World Association of Sarcoidosis and Other Granulomatous disease (ERS/WASOG) guidelines for the diagnosis of sarcoidosis. Histological confirmation of non-caseating granulomas was obtained by flexible bronchoscopy, as well as bronchoalveolar lavage (BAL) fluid for the determination of BAL CD4+/CD8+ ratio. Patients with other known systemic inflammatory illnesses were excluded, as well as patients with acute infection, patients on immunosuppressive drugs or immunotherapy and patients with active cancer”. Please add more information in the text regarding the recruited patients (age, sex, onset of pathology, years of pathology)

5.         Conclusion 351-354:” Out of the typical HRCT characteristics of intrathoracic sarcoidosis, hilar and mediastinal lymph node enlargement, 351 perilymphatic/peribronchovascular involvement, predominantly ground glass lesions and pulmonary fibrosis exhibit 352 underlying differences in the measured serum markers, especially SAA, CA15.3 and SP-D. Further studies will be 353 needed to establish the prognostic value and possible clinical implications of patient stratification according to our 354 proposed HRCT disease patterns.” Please re-write more incidive conclusion.

6.         Please add any strengths and weaknesses of the study

Reviewer 2 Report

The paper is interesting and well-organized; however, they have some comments:

- the study population is heterogeneous. several stage II sarcoidosis.

- It was not reported the definite diagnosis of ILD.

- serum samples were centrifuged at 3000 × G . Are you sure it was "G" and not "rpm"? If it was 3000 rpm, please covert it in "g".

- on what basis do they decide whether to keep it at -20 or -80 degrees?

- Was there an increase of KL-6 concentrations according to scadding stages?

Although the authors determined a cut-off value for CTO, were CTO values compared with healthy or other interstitial lung diseases?

Minor editing of English language required

Author Response

Point 1:

The study population is heterogeneous. several stage II sarcoidosis.

Response 1:

We thank the reviewer for the comment and have included an explanation in the section on Study strengths and weaknesses, as follows:

“Our patient group features an uneven distribution of patients according to Scadding stage. Scadding stage 0 was purposely omitted, as we focused on patients with confirmed intrathoracic disease. While we would expect most patients (45-65%) to be in Scadding stage 1, this number is much lower in our patient group, only 11,5%, with 79,7% of patients in stage 2 and 7,6% in stage 3 [9]. In patients with with stage 1 sarcoidosis that were referred to our sarcoidosis centre and presented with Löfgren syndrome, HRCT and histopathologic confirmation of diagnosis were only rarely performed, due to the diagnostic features of Löfgren syndrome and the mostly benign disease course [6], [17]. In contrast, both HRCT and histopathologic confirmation of disease by bronchoscopy were needed for inclusion in our study. Additionaly, some patients with Löfgren syndrome, as a form of stage 1 sarcoidosis, were diagnosed by specialists in rheumatology or dermatology, who also initiated treatment before referral, again precluding their inclusion in our study, which only enrolled treatment-naïve patients.”

Point 2:

It was not reported the definite diagnosis of ILD.

Response 2:

Thank you for your comment and we apologise for the omission. Indeed, we have now addressed this issue in the first paragraph of ”Results” as follows:

“Based on multidisciplinary review and clinical, radiological and histological correlation according to ERS/WASOG [4], [6], 55 patients were determined as having definite diagnosis of sarcoidosis interstitial lung disease.“

Point 3:

Serum samples were centrifuged at 3000 × G . Are you sure it was "G" and not "rpm"? If it was 3000 rpm, please covert it in "g".

Response 3:

We thank the reviewer for the comment and apologize for this mistake. Indeed it was 3000 rpm and 10 min. We corrected the sentence as follows:

“Serum samples were collected after clotting of peripheral venous blood and centrifuged at 1800g for ten minutes, aliquoted and stored short term at -20 °C and long term at -80 °C until used for further analysis.”

Point 4:

On what basis do they decide whether to keep it at -20 or -80 degrees?

Response 4: We apologize the reviewer for unclear sentence. We corrected the sentence (above) to make it more clear.

Point 5:

Was there an increase of KL-6 concentrations according to scadding stages?

Response 5:

We thank the reviewer for the question. In our cohort, sarcoidosis patients have on average statistically significantly larger CA15.3 values compared to healthy subjects. “Among patients, the Scadding stage is not significantly correlated with CA15.3 (correlation coeficient 0.13, p value 0.26).” This sentence was added to the Discussion section.

Point 6:

Although the authors determined a cut-off value for CTO, were CTO values compared with healthy or other interstitial lung diseases?

Response 6:

We thank the reviewer for the comment and have offered a further explanation in the Discussion, as follows:

“Although we have not directly compared CTO activity in sarcoidosis patients and healthy controls in this study, results are in accordance with previously published research performed by our sarcoidosis group in the same population where direct comparison of patients and healthy controls showed significantly increased CTO activity in sarcoidosis patients (p<0.001), confirming the high sensitivity of this disease marker [18]–[20]. CTO activity correlated well with disease activity and severity and was confirmed to be usefull in patient follow-up and detection of disease relapse [21], [22]. Despite having good specificity, activity above controls was also found in patients with asbestosis, non-sarcoidosis pulmonary fibrosis and lung cancer [18].”

Round 2

Reviewer 2 Report

The authors exhaustively replied to the comments.

Minor editing of English language required